# A Speech Recognition Model Building Method Combined Dynamic Convolution and Multi-Head Self-Attention Mechanism

Wei Liu , Jiaming Sun, Yiming Sun and Chunyi Chen *

College of Computer Science and Technology, Changchun University of Science and Technology, Changchun 130022, China; liuwei@cust.edu.cn (W.L.); jiamingsun0721@gmail.com (J.S.); sunyiming@cust.edu.cn (Y.S.)
* Correspondence: custchency@126.com; Tel.: +86-13504326248

**Abstract:** The Conformer enhanced Transformer by using convolution serial connected to the multi-head self-attention (MHSA). The method strengthened the local attention calculation and obtained a better effect in auto speech recognition. This paper proposes a hybrid attention mechanism which combines the dynamic convolution CNNs and multi-head self-attention. This study focuses on generating local attention by embedding DY-CNNs in MHSA, followed by parallel computation of the globe and local attention inside the attention layer. Finally, concatenate the result of global and local attention to the output. In the experiments, we use the Aishell-1 (178 hours) Chinese database for training. In the testing folder dev/test, 4.5%/4.8% CER was obtained. The proposed method shows better performance in computation speed and the number of experimental parameters. The results are extremely close to the best result (4.4%/4.7%) of the Conformer.

**Keywords:** speech recognition; attention; dynamic convolution; transformer

## 1. Introduction

With the transformer [1] proposed in the field of speech recognition, there are more and more studies based on this model in the end-to-end continuous speech recognition task. Conformer is a popular speech recognition model that is improved based on Transformer [2]. Transformer is more effective in extracting long sequence dependencies, while convolution is good at extracting local features [3]. The main improvement in Conformer is that convolution is applied to Transformer, improving the effect of the model on long-term sequences and local features [4].

In particular, the Conformer enhances the local features of Transformer by convolution obtaining better recognition results [5] than Transformer. Compared with RNN, Transformer shows the advantages of faster computing speed and better global information representation in the fields of speech recognition, machine translation and natural language processing [6–8]. In recent years regarding end-to-end continuous speech recognition, the best results basically use the transformer structure.

Regarding continuous speech recognition tasks, excluding the speech feature itself, there is a major problem in the correlation of time series and uncertainty of speech duration. The Transformer model can effectively deal with both of these issues. The multi-head attention structure can extract the dependencies between contexts in time series. The model's Encoder and Decoder length can also deal with the uncertain speech duration problem for input and output length.

In the wide range of applications for sequence task processing, although the Transformer model has proved to be efficient, the model also bears certain problems, such as global information redundancy. In continuous speech recognition, when facing a long sequence problem, although context dependency is needed, the context does not require a

very large range to know the global state when using the attention mechanism. In some cases, the effect is better with only information of multiple time steps before and after, rather than the global information. In the case of retaining global information, the key problem to be solved by the model is in determining how to focus attention on the important information. Based on this idea, we proposed a new construction method of a mixed attention model to obtain local attention while retaining global attention. Finally, we proved the effectiveness of the model structure through experiments.

Regarding studies on the local and global attention of the Transformer model, many researchers have offered different solutions. The Long Short Range Attention (LSRA) [9] method uses a dual-branch structure to parallel calculate local attention and global attention. First, it uses MHSA to calculate global attention. Then, it employs the combination module including linear layer and CNN to calculate local attention. Finally, it integrates these in the feed forward network module (FFN) layer, which obtains satisfactory results in many sequence tasks. The recently proposed Conformer structure [10] adds a series of CNN combination modules to extract local information from global attention after the completion of MHSA calculation. This structure ensures the dependence between local information when the global attention remains intact. The Conformer structure achieves optimal results in speech recognition tasks on multiple datasets [11].

Under the condition of retaining MHSA, the current mainstream practice is to use CNN to obtain local attention. CNN is proficient in extracting fine-grained information from the feature matrix [12]. However, the method is not advantageous in extracting long context-dependent information from the sequences. Therefore, combining MHSA and CNN can not only retain context dependence, but also obtain local attention. They appear to form a global and local complementary relationship. MHSA reflects the relationship between a certain time step and the overall time step, but the MHSA mechanism focuses excessively on the global information. This leads to each output vector carrying global information. In order to ensure the effectiveness of the global information, the before and after information of hundreds of frames are processed. The processing leads to poor information conversion effect on fine-grained structures.

If we do not consider the role of grammar, the processing of over a dozen frames of information is sufficient regarding continuous speech recognition sequences. Even with consideration of the full context information, the left and right extension of dozens of frames can ensure sufficient recognition. In addition, the scope of context information has different effects on different text, thus the global information processing should not be fixed. Excessive dependence on global relations to make the final result judgment will lead to some important local information being diluted or even submerged by global relations.

The popular Conformer model is essentially based on the modified Transformer model. The purpose of improvement is to solve the problem of mutual exclusion between local information and global information. The core module of Conformer essentially uses convolution after attention, and then uses attention to extract local features. It is a classic 'sandwich' structure, also known as the Macaron structure. The disadvantage of this approach is the serial processing which reduces efficiency. The CNN module must wait for one layer to finish before processing another layer. A total of three layers are needed to complete the final feature extraction.

Focusing on the above problems, the main idea of the proposed method firstly uses convolution within the MHSA in parallel, and then combines the convolution results with the attention layer to obtain the new feature. The convolution process is directly embedded into the attention layer, and then processed in parallel. The global attention and local attention are processed by in only one step. The proposed method of parallel convolution inside MHSA is better and more efficient than the end process operation in Conformer.

Dynamic convolution can adaptively fuse multiple convolution kernels according to input [13]. The multiple convolution kernels make the convolution results adaptive. Compared with static convolution, dynamic convolution can significantly improve the expression ability and performance of the model. Dynamic convolution does not use one

convolution kernel at each layer, but dynamically gathers together according to the attention of multiple parallel convolution kernels. These convolution kernels are dependent on input, and the kernel bears a small size and high computational efficiency. These dynamic convolution kernels are aggregated in a nonlinear manner, that is, different attention is given to obtain different information expression ability so as to better represent the difference between local and global information.

Dynamic convolution aims to dynamically aggregate multiple parallel convolution kernels, without increasing the depth and width of the network. It adaptively selects different convolution parameters according to the different input characteristics. Finally, it again convolutes the fused features. Figure 1 describes the calculation process of dynamic convolution. It is assumed that m convolution kernels, i.e., $\{conv_m\}$, are used for convolution. The input features will extract the local features through the average pooling layer and the linear layer. Then, the weight of m convolution kernels is computed (i.e., convolution kernel attention) $\pi_m$, at the current time step is obtained. Its properties are as follows.

$$0 \leq \pi_k \leq 1, \sum_{k=1}^{m} \pi_k = 1 \tag{1}$$

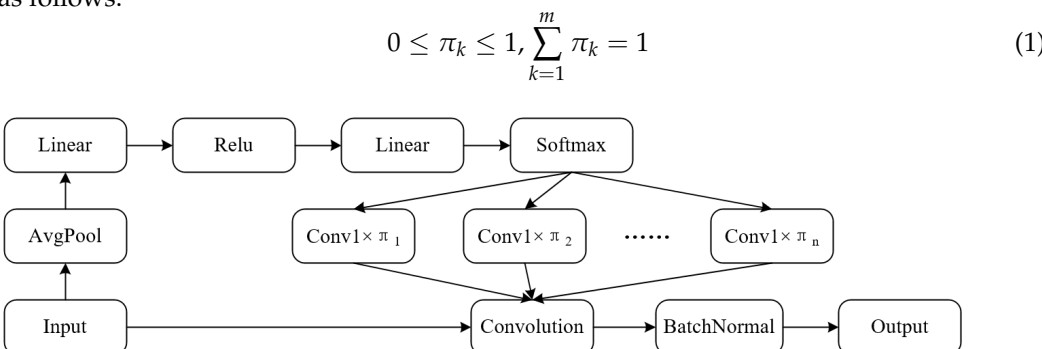

**Figure 1.** Introduction and calculation process of DY-CNNs.

The convolution Kernel Convn$'$ with new convolution input characteristics can be obtained by combining the weight of m convolution kernel with m convolution kernels in the convolution layer.

$$Conv'_n = \sum_{i=1}^{m} conv_i * \pi_i \tag{2}$$

Figure 1 shows the weight of the convolution operation obtained by a series of operations. Next, is the convolution weight with its corresponding convolution kernel. Finally, the final output feature is obtained by a batch normalization layer. The operation of convolution layer is to obtain different convolution kernels at different time steps. Then it uses convolution layer to extract local features. The context information for the nth time step $c_n = Conv'_n(x_{n-d \sim n+d})$ is obtained by the above operation. The convolution kernel attention is calculated for each time step in the feature matrix X. The convolution kernel is fused to obtain the convolution kernel $Conv'_1$, $Conv'_2$, $\ldots, Conv'_n, \ldots, Conv'_t$ corresponding to each time step. Using these convolution kernels to make convolution operations on each time step corresponding the feature matrix X, the feature matrix $C = \{c_1, c_2, \ldots, c_n, \ldots, c_d\}$ containing all the context range d can be obtained.

The convolution kernel size and pooling kernel size in DY-CNNs are dynamically determined by the context information length. For obtaining speech feature information $c_n$ of the nth time step context d in matrix $X = \{x_1, x_2, \ldots, x_n, \ldots, x_t\}$. In the X matrix, calculate the convolution of the internal characteristics whose range is $x_{n-d \sim n+d} = \{x_{n-d}, \ldots x_n, \ldots x_{n+d}\}$. The convolution kernel size is $(2d + 1, 1)$. The convolution kernel size is also the average pooling kernel size when obtaining the attention of the convolution kernel. The path of the corresponding operation process should also be consistent in the convolution and pooling. As shown in Figure 2, the process of pooling and convolution is carried out when the context range d is 2 and the convolution kernel size is $(2d + 1, 1)$.

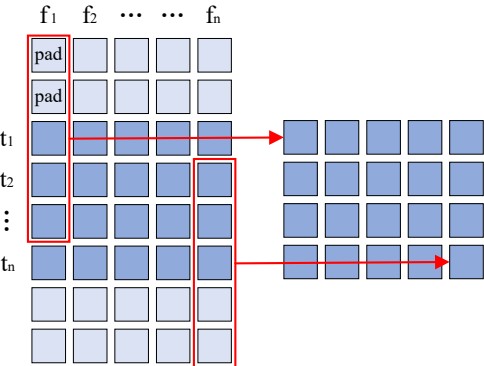

**Figure 2.** Pooling and convolution process of DY-CNNs.

In the diagram, the pooling kernel and convolution kernel are 5. It adds two time steps to make up 0 in the up and down directions. Then, the process of pooling and convolution is implemented. The purpose of this step is to make the size of the matrix after pooling and convolution equal to that of the input matrix before padding. Each element of the output matrix represents the integration result of the context information at the location of the input matrix. The average pooling output reflects the overall performance of the feature for a period of time. The convolution output represents the representation of the current time step with context information.

Through the above analysis, it can be seen that the dynamic convolution process can better extract the local attention information according to the context characteristics. The proposed method is to use MHSA to generate global attention and DY-CNNs to generate local attention inside the attention layer. Finally, the results of the two kinds of attention are concatenated and output to the next layer, and the global and local attention mechanism network structure based on dynamic convolution is obtained.

## 2. Materials and Methods

### 2.1. The Structure of Multi-Attention

When calculating global attention, we use the original structure of MHSA in Transformer, because it can effectively solve the problem of obtaining global information in the sequence. Then, the sine position coding is used to obtain the absolute position information of the sequence. Suppose a given input feature $X \in \mathbb{R}^{T \times d_k}$, where $d_k$ is the dimension of feature vector and T is the total number of time steps. MHSA inputs feature X through three linear layers and converts it into Q, K and V. Where $Q, K \in \mathbb{R}^{T \times d_k}$, $V \in \mathbb{R}^{T \times d_v}$. The $d_v$ represents the dimension of the matrix V. Matrix Q represents the state of the global time step. Matrix K represents the state of each time step to be matched. In addition, matrix V represents the state of the current time step waiting for the given weight. The three can calculate the attention of a moment by the following formula.

$$Attention(Q, K, V) = softmax\left(\frac{QK^T}{\sqrt{d_k}}\right)V \tag{3}$$

The score calculation of matrix Q and K is the weight matrix representing the relationship between each time step and the global time step. Matrix V needs to know the relationship between the current time step and the global time step. The attention calculated by this formula is the attention with global information [14].

In MHSA, the matrix Q, K and V are divided into h-orders. Some corresponding matrices Q, K and V are obtained. Each group of corresponding Q, K and V is called the head. The attention of each head is calculated, respectively. Finally, the final attention result is obtained by concatenation.

$$MultiHead(Q, K, V) = Concat(head_1, \ldots, head_h)W^O \tag{4}$$

$$head_i = Attention\left(QW_i^Q, KW_i^K, VW_i^V\right) \tag{5}$$

$W_i^Q \in \mathbb{R}^{d_{model} \times d_k}$, $W_i^k \in \mathbb{R}^{d_{model} \times d_k}$, $W_i^v \in \mathbb{R}^{d_{model} \times d_v}$, $W^O \in \mathbb{R}^{hd_v \times d_{model}}$, $h$ is the head number, $d_{model}$ is model dimension and $d_k$ is the dimension of matrix key.

### 2.2. Feed Forward Network Module

When connecting the attention block, two feed forward neural network modules (FFN) [10,15] are used. Each FFN contributes half value and the model effect can be slightly improved by experiments. A single FFN is composed of layer normalization, linear layer, swish activation function and dropout. The normalization layer makes the input features stable. The swish function enables FFN to fuse with the model faster. The results obtained by the module will be output to the next layer in a semi-step residual method. Compared with without semi-step residual method, this connection method has more advantages in the training process. The given feature input $x_i \in \mathbb{R}^d$ is passed through the feed forward network. The result $\widetilde{x}_i$ is calculated by the following formula.

$$\widetilde{x}_i = x_i + \frac{1}{2}\text{FFN}(x_i) \tag{6}$$

The model uses two FNNs to clamp the proposed attention method in the middle. The structure still belongs to the Macaron feedforward module. Each FFN is connected to the next layer in a semi-step residual manner. The general design structure is shown in Figure 3. Experiments show that the FFN structure will increase the parameters, but it can improve the information transmission and learning efficiency of attention compared with the single FFN structure.

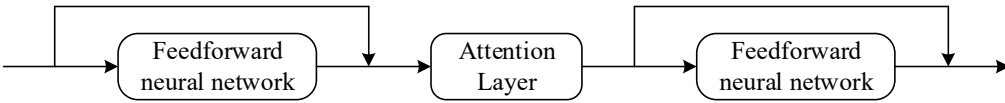

**Figure 3.** Structure diagram for each block.

### 2.3. The Structure of Dynamic Convolution Attention

The multi-attention mechanism generates three matrices Q, K and V in calculation. In the calculation of local attention, the matrix Q is used as the input $\pi_l$ to generate the convolution kernel weight of DY-CNNs. Then this weight is assigned to DY-CNNs multiple convolution kernels $\{conv_l\}$. The convolution kernel $Conv'_n$ corresponding to the context information of step n is generated by fusion, to perform convolution operation on the matrix K. The result assigns local weights to the matrix V, that is, the correlation of context information $c_n$. Then this score is assigned to the matrix V again by convolution. Finally, the local attention corresponding to each time step in the matrix V is obtained.

Different from other studies, we improved the attention within the structure of multi-head attention. First, we obtain local attention using dynamic convolution. Then, we concatenate local attention and global attention from the proposed structure. Finally, we explore the effect of attention. The matrices Q, K, and V are recalculated. The matrix S of local attention with local information is generated. The matrices Q and K are obtained from the linear layer of multiple attentions.

Assuming that the size of the feature matrix is (T, F), which is obtained from the matrix Q, K and V. The hyper parameter $d$ is defined, which represents the feature of looking forward and backward at $d$ time steps respectively from the current time step. The information in $2d + 1$ scope is aggregated (equivalent to the spelling frame operation). Symbol m is the number of convolution kernels of DY-CNNs, that is, how many convolution kernels are fused. The overall attention calculation process of DY-CNNs is shown in Figure 4. When the matrix Q after padding is added into the average pooling layer, matrix Q′ with the same size as the matrix Q will be obtained. The size of the pooling kernel of the average pooling layer is $(2d + 1)$, which is the same as that of the convolution kernel. The input

size is transformed into (T, F/4) through a linear layer. Finally, $\pi_m$ with dimension (T, l) is obtained by linear layer and Softmax activation function. $\pi_m$ means that each time step in T is obtained by $m$ weights for the convolution kernel of the current time step. The convolution kernel of DY-CNNs is obtained by fusing $\{\pi_m\}$ and $\{conv_m\}$. Finally, the weight S of local attention is obtained by grouping convolution along the previous pooling path.

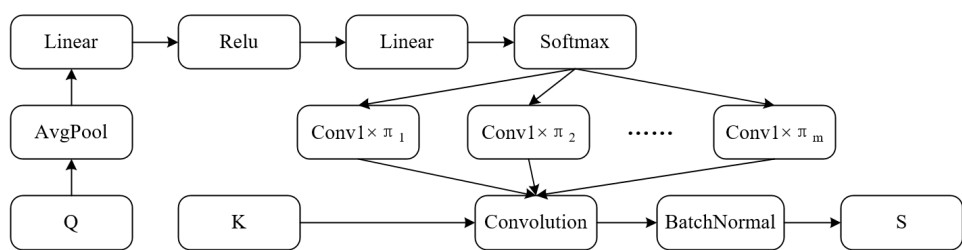

**Figure 4.** Flowchart of DY-CNNs attention.

Unlike dynamic convolution, the convolution object and the object that obtain the convolution kernel weight are not from the same input feature. Here we use the matrix Q to obtain the attention of the convolution kernel, and then make convolution with the matrix K. The convolution kernel of matrix S and V is $1 \times 1 \times 2$, and the local attention is obtained with the size of (T, F). Local attention will concatenate with the global attention obtained in MHSA, and the attention matrix with dimension (T, 2×F) is obtained. Finally, a linear layer is passed to reduce the size of the feature to (T, F).

$$S = DyConv(Q, K) \tag{7}$$

$$Local\ Attention = LConv(S, V) \tag{8}$$

$$utput = Linear(Concat(Global\ Attention, LocalAttention)) \tag{9}$$

In Algorithm 1, the pseudo code is used to illustrate how dynamic convolution is used to realize the parallel calculation of attention.

---

**Algorithm 1.** Dynamic convolution attention

---

**Input**: A feature matrix $X$, the length of $X$ is $n$
**Output**: Attention matrix $S$
**Initialization**: $i = 0$, $m$ is the number of kernel for dynamic covolution, $conv = \{c_1, c_2, \ldots, c_m\}$ is the parameters of dynamic covolution

1 : $Q = W_q X$, $K = W_k X$, $V = W_v X$; $//W_q, W_k, W_v$ are the parameters of linear layers
2 : Compute the global attention $G$ according to Equation (3);
3 : Pad 0 to the beginning and end of $Q, K$, then cut them into $n$ parts
$Q' = \{q_1, q_2 \ldots, q_n\}$, $K' = \{k_1, k_2, \ldots, k_n\}$;
// the length of $q_k$ is the scope of local attention
4: **while** i < n do
5 : $p_i = AveragePool(q_i)$;
$//The window size of Average Pool is the scope of local attention$
6 : $h_i = W_2 ReLu(W_1 p_i)$, $len(h_i) = m$;
$//W_1, W_2$ are the parameters of linear layers, $h_i = \{\pi_1, \pi_2, \ldots, \pi_m\}$
7 : $Conv_i = h_i conv = \sum\limits_{j=1}^{m} \pi_j c_j$;
$//Conv_i$ is the convolution kernel of current time step
i8 : Convolute $p_i$ to $z_i$ with convolution kernel $Conv_i$;
9 : $Z.append(z_i)$;
10 : $L = BatchNorm(Z)$ $//L$ is the local attention;
11 : Concatenate $G$ with $L$ and convolute it to $S$; $//$The size of $S$ is the same as $X$
12 : return $S$

---

### 2.4. Parameter Reduction

Increasing the dynamic convolution attention within the multi-attention will increase the parameters of the model, although the increase is not large. During the experiment, we attempted to reduce the model parameters while keeping the final model accuracy unchanged. When the input features are input into the linear layer to obtain the matrices Q, K and V, the original dimension F is reduced to half, that is, the dimensions of $d_Q$, $d_K$ and $d_v$ are changed into F/2. When the head remains unchanged in the multi-head attention, the global attention dimension is (T, F/2), and the local attention dimension after dynamic convolution is (T, F/2). The size of the output obtained by concatenating the two is (T, F), and it is finally mapped to the dimension of (T, F) through a linear layer. Experiments show that the parameter reduction CTC method can reduce the decoding parameters by 2.17M when block = 12. The method can also achieve better in the same epoch of training time.

### 3. Experiment and Result Analysis

#### 3.1. Data Introduction

Aishell-1 [16] Chinese 178-h open-source speech database is used to evaluate the proposed model. The data are recorded by three different devices for 400 speakers from different accent areas in China. We also use AIDATATANG and HKUST open-source dataset to verify the effectiveness of proposed method on different condition. AIDATATANG is a 200-h open Chinese Mandarin telephone voice library provided by Beijing Data Technology Co., Ltd. Bejing City, China. HKUST is a 200-h Chinese telephone data set. The dataset uses ESPNet and Kaldi as open-source data.

The feature is obtained by 80-dimensional logarithmic Mel spectral coefficient, 25 ms window size and 10 ms displacement. During the training process, the data were transformed by velocity. The velocity variation [17] coefficients were operated according to 0.9, 1.0 and 1.1, and the spectral enhancement [18] was performed. The window size of time warping was 5. The size of random shielding in frequency domain was 30. The size of random shielding in time domain was 40. The data were normalized by global CMVN [19]. The Kaldi-like script is used for data processing in the data preparation phase.

#### 3.2. Training and Results

The lightweight ESPNet [20,21] is used as the experimental platform. After feature extraction, the convolution front-end module in ESPNet is used for downsampling to a quarter of the original. The same structure and parameters as Transformer are adopted at the encoder. The number of blocks is 12. The size of each output time step is 256. The number of attention heads is 4. The number of hidden layer nodes of the neural network in the attention layer is 2048 and the parameters are reduced in each layer. The Macaron network is used to connect attention outside the attention layer, and the activation function in the linear layer is the swish function. By parameter reduction, the 256-dimensional is reduced to 128-dimensional and the attention is calculated. Four convolution kernels are used to fuse the convolution kernels that generate dynamic convolutions. The convolution kernel size is set to 15. Finally, the output of the multi-head attention is concatenated. The output is carried out in a linear layer with 256 dimensions. A transformer decoder with 6 blocks, 2048 linear units and 4 attention heads is used at the decoder. During training and decoding, [22] the CTC attention combination method was used [23]. Encoder and decoder used cross entropy function and label smoothing with weight of 0.1. CTC loss function is used at encoder side for auxiliary alignment to help the model learn more alignment information.

The model is iteratively optimized through the connection layer in the training process. In the training stage, the Adam optimizer is used to optimize the model. $\beta_1$ is set to 0.9, $\beta_2$ is set to 0.98 and $\varepsilon = 10^{-9}$. The learning rate schedule of transformer is used to linearly increase the learning rate in the training, and the initial learning rate is $5 \times 10^{-4}$. The inverse square root of the number of steps is reduced in proportion and 60 epochs are trained in the state of warmup_steps = 30,000. The language model of transformer structure

is trained by scripts in the training set, and words are used as the modeling unit. The language model sets the feature embedding to 128 dimensions, layer to 8, Head to 8, linear unit to 2048, attention output to 512, dropout of FFN to 0.1, and trains 15 periods. Finally, the language model that performs best on the dev set is added to the decoding in a shallow fusion manner.

Set ctc_weight = 0.6, lm_weight = 0.3 when reasoning. The model is decoded by beam search with a width of 20, and the parameters of the 10 best models on the dev set are averaged as the final model.

In the training process, firstly add the <sos> and <eos> symbols at the beginning and end of the text information. Then the full sequence of feature matrix and text information is fed to the encoder and decoder, respectively, and the loss between the output of the decoder and the real label is continuously optimized through the back-propagation algorithm to optimize the model. When decoding, the feature matrix is completely input into the encoder and the attention matrix is output. The previous output character is continuously input on the decoder side, and is then combined with the attention matrix to output the next candidate character. If the decoder has not output any character, replace the input with <sos>. The decoding of the model is an autoregressive process, and the loop stops until <eos> is encountered.

This section may be divided by subheadings. It should provide a concise and precise description of the experimental results, their interpretation, as well as the experimental conclusions that can be drawn.

### 3.3. Comparison of Results

When the convolution kernel size is set to 15 inside the dynamic convolution, the trained model obtains a very competitive result in the Aishell-1 dataset. Without using the relative position coding, but using the language model, the result is 4.7/5.0 in the dev/test directory. After downsampling, the word error rate was reduced to 4.5/4.8 by using relative position coding, which was basically consistent with that of Conformer. In the experiment, the trained model is also compared with other existing models from multiple dimensions, including Hybrid Model, Transformer, LAS, Conformer results (CER) as follows. In the existing models, the proposed model obtains more competitive results without using relative position coding, and the number of parameters is also guaranteed.

In Table 1, we compared different models in multi-dimension on Aishell-1 dataset. The baseline is the conventional Transformer method. Compared with this method, the proposed method improved 1.3/1.7 in dev/test. If using relative position coding, the results are very similar with Conformer. However, the parameter scale is about 4M smaller than Conformer.

**Table 1.** Comparison of experiments for different models in multi-dimension on Aishell-1.

| Method | E2E | LM | Params | Dev | Test |
|---|---|---|---|---|---|
| Baseline [6] | Y | Y | 17.62 | 6.0 | 6.7 |
| TDNN-LFMMI [24] | N | Y | - | 6.44 | 7.62 |
| LAS [25] | Y | Y | - | - | 8.71 |
| Conformer(M) [10] | Y | Y | 33.47 | 4.4 | 4.7 |
| Proposed method | Y | Y | 29.66 | 4.7 | 5.0 |
| Proposed method + relative position coding | Y | Y | 29.66 | 4.5 | 4.8 |

In order to verify the universality of the proposed model, a comparison of the experiments was also performed on other Mandarin datasets named Aishell-1, AIDATATANG and HKUST. The data time of the three datasets is about 200 h. As shown in Table 2, the proposed method still exhibits the best results over the baseline system on these datasets.

**Table 2.** Comparison of experiments on different datasets.

| Dataset | Test Sets | Baseline | | Proposed Method | |
|---|---|---|---|---|---|
| Aishell-1 | dev/test | 6.0 | 6.7 | 4.5 | 4.8 |
| AIDATATANG | dev/test | 5.9 | 6.7 | 4.4 | 4.9 |
| HKUST | dev | 23.5 | | 22.4 | |

Table 3 shows the comparison of experiments for training time. The proposed method is tested on three training sets. Compared with Baseline, although the time increases, it can be seen from Table 1 that the accuracy is improved by 1.3/1.7 on dev/test. Compared with Conformer, the training time is reduced slightly while the accuracy is basically unchanged. With the increase of training data, time will be further reduced.

**Table 3.** Comparison of experiments for training time.

| DataSet | Model | Time |
|---|---|---|
| Aishell (178 h) | Baseline | 1 day 13 h |
| | Conformer | 2 day 8 h |
| | Proposed method | 2 day 5 h |
| AIDATATANG (200 h) | Baseline | 1 day 16 h |
| | Conformer | 2 day 14 h |
| | Proposed method | 2 day 8 h |
| HKUST (200 h) | Baseline | 1 day 15 h |
| | Conformer | 2 day 14 h |
| | Proposed method | 2 day 13 h |

In order to explore whether dynamic convolution and parameter reduction have a positive effect on the results, the ablation experiment is further carried out. Firstly, the parameter reduction in the model is removed. It can be concluded from the experiment that parameter reduction can reduce the parameters of the model and has little effect on the model results.

We then remove the dynamic convolution combined with attention in the model structure, and use MHSA to perform experiments. The results show that the word error rate on the development set increases by 0.3, and the word error rate on the test set increases by 0.4. The overall number of parameters increased by about 1.6M compared with the traditional MHSA model. It can be seen that the results can be improved at a relatively small cost by increasing the attention of the dynamic convolution combination. The results shows that the local attention information extracted by the proposed method can effectively help the model to learn some relatively fuzzy information under global attention.

To identify better attention results, we also tested different structure by placing CNN in different positions. If the part of DY-CNN in Figure 5 is replaced by CNN, in Table 4, named the replacement is the method with CNN. If we place CNN at the end of the structure as with the Conformer, in Table 4, named the structure is the method with rear CNN. The results show that the proposed method in Figure 5 has the best results.

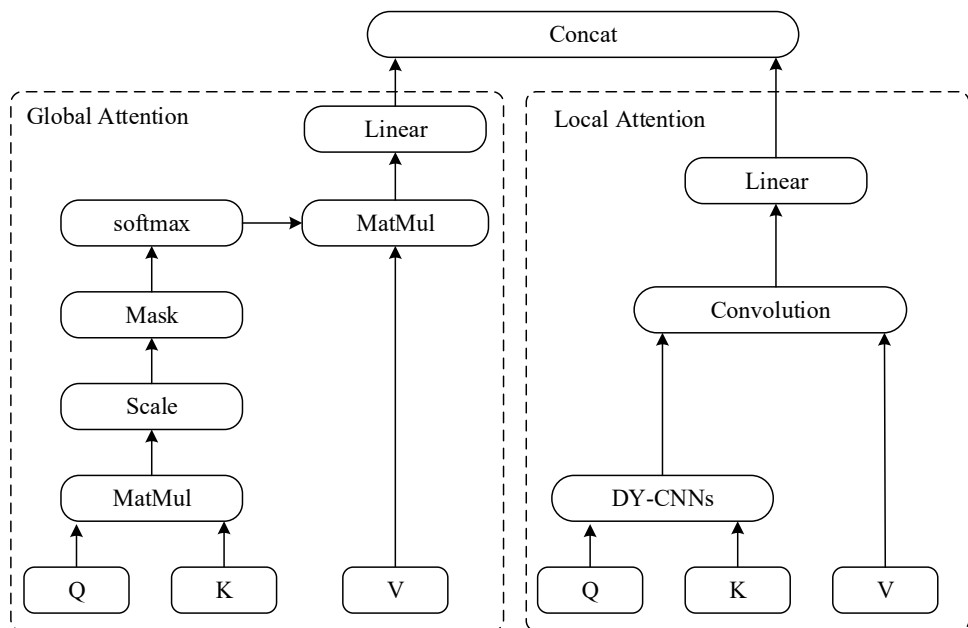

**Figure 5.** Left part shows the global attention structure, right part shows the local attention structure, and both inputs are from the same matrices of Q, K and V.

**Table 4.** Comparison of different model structure.

| Model Structure | Params | Dev | Test |
|---|---|---|---|
| Proposed method | 31.83 | 4.7 | 5.0 |
| Method with CNN | 27.09 | 5.4 | 5.7 |
| Method with rear CNN | 27.09 | 5.3 | 5.5 |

In order to verify the effectiveness of dynamic convolution in the model, we also replaced the dynamic convolution in the model with a common convolution operator for comparative experiments. The experimental results show that dynamic convolution can significantly improve the ability of the model. Compared with the method without DY-CNNs, the results improved 0.5/0.4 in dev/test. Compared with the method with CNN, the results both improved 0.7 in dev/test. Figure 6 shows position of the parameter reduction. Table 5 demonstrates the recognition accuracy with parameter reduction.

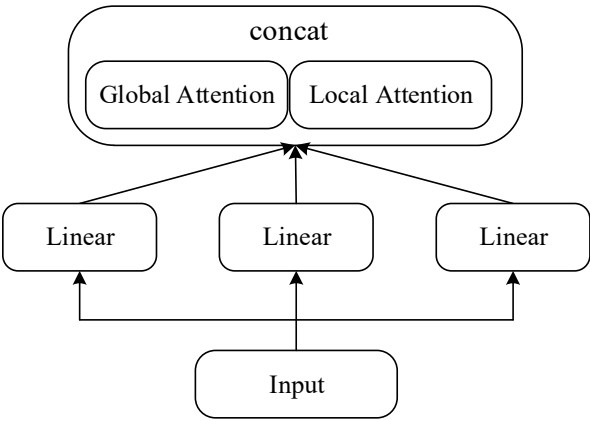

**Figure 6.** Parameter reduction diagram.

**Table 5.** Experiment results of parameter reduction.

| Model Structure | Params | Dev | Test |
|---|---|---|---|
| Proposed method + relative position coding + Parameter reduction | 29.66 | 4.5 | 4.8 |
| Proposed method + relative position coding + Parameter reduction(without DY-CNNs) | 28.53 | 5.3 | 5.5 |
| Proposed method + Parameter reduction | 29.66 | 4.7 | 5.0 |
| Proposed method + Parameter reduction (without DY-CNNs) | 28.53 | 5.3 | 5.5 |
| Proposed method | 31.83 | 4.7 | 5.0 |
| Proposed method(without DY-CNNs) | 30.24 | 5.2 | 5.4 |
| Method with CNN | 27.09 | 5.4 | 5.7 |

At the same time, we focus on the kernel size of convolution in dynamic convolution. The size of the convolution kernel is a very important parameter. For the whole model, it represents the size of the field of view to obtain context information, which will directly affect half of the features in the attention module. The sounding time of a Mandarin character is about 600 ms. The experiment selects 5 (200 ms), 15 (600 ms), and 25 (1000 ms) for the convolution kernel size. A comparative experiment is conducted under the same training environment.

The results are shown in Table 6. It can be seen when the convolution and kernel is 15 (look forward 7 time steps and look back 7 time steps), the model obtains a relatively low word error rate. The word error rate increases when the convolution kernel size is 5 or 25. Therefore, for the proposed model, it is more appropriate to set the convolution kernel size to 15.

**Table 6.** Influence of different kernel size on experiment results.

| Kernel Size | Dev | Test |
|---|---|---|
| 5 | 4.8 | 5.1 |
| 15 | 4.5 | 4.8 |
| 25 | 4.8 | 5.1 |

## 4. Conclusions

This article is based on the AED (Attention-based Encoder–Decoder) model structure, and improvement of the module of multi-head self-attention mechanism in Transformer. We propose an end-to-end speech recognition model based on hybrid attention mechanism. The dynamic convolution method is first introduced in the field of speech recognition, to solve some problems of the Transformer model concerning recognition accuracy. At the same time, the parameter reduction of the model is optimized to reduce the calculation burden as much as possible caused by the increase of parameters.

We firstly discussed the application method of dynamic convolution combined with original attention in continuous speech recognition. Then we proposed an optimized mixed attention mechanism. Aiming at the problem of balanced distribution of local attention and global attention in long sequences, the proposed method enhances the correlation between local attention and global attention by introducing dynamic weighting, a change from Serial Computing Structure to Parallel Computing Structure, increased efficiency of

operations, and a reasonable reduction of the parameters of the model through experiments. Finally, through multi-dimensional experiments, comparing the results of this new dynamic convolution with that of traditional methods, the results prove its effectiveness in long sequence tasks of continuous speech recognition. The proposed model obtains a more competitive result at the cost of a relatively small parameter increase.

There are many unreasonable aspects of the currently used AED model. For example, streaming speech recognition is not supported. The complexity of the auto-regression process is high in the decoding part. The method is unable to use prior textual information, etc. Although the FNN structure used in the article significantly improves the recognition results, it also greatly increases the model parameters and computational overhead. The next step is to further optimize the long-sequence speech modeling features from the perspective of feature calculation.

**Author Contributions:** Writing—original draft preparation, W.L.; resources, J.S.; writing—review and editing, Y.S.; supervision, C.C. All authors have read and agreed to the published version of the manuscript.

**Funding:** This research was funded by the NATIONAL NATURAL SCIENCE FUND PROJECT, grant number U19A2063 and JILIN PROVINCIAL DEPARTMENT OF EDUCATION SCIENCE AND TECHNOLOGY RESEARCH PLANNING PROJECT, grant number JJKH20220779KJ.

**Institutional Review Board Statement:** Not applicable.

**Informed Consent Statement:** Not applicable.

**Data Availability Statement:** Datasets are available at http://www.openslr.org/33/ (accessed on 16 May 2022) http://www.openslr.org/62/ (accessed on 16 May 2022) https://catalog.ldc.upenn.edu/LDC2005S15 (accessed on 16 May 2022).

**Conflicts of Interest:** The authors declare no conflict of interest.

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
