# Peer review of "A Speech Recognition Model Building Method Combined Dynamic Convolution and Multi-Head Self-Attention Mechanism"

_electronics, doi:10.3390/electronics11101656_

Round 1
Reviewer 1 Report
overall contents are improved
I am happy with the responses from the author
Reviewer 2 Report
The paper presents a generation of local attention by embedding DY-CNNs in MHSA. This paper is a resubmission, and the faults in the last version were improved, and it can be accepted after a linguistic revision.
This manuscript is a resubmission of an earlier submission. The following is a list of the peer review reports and author responses from that submission.
Round 1
Reviewer 1 Report
The paper presents a dynamic convolution idea that can be integrated with well-known transformer architecture for speech recognition. The dynamic convolution is found here to help improve overall accuracy. Only one single dataset is used to verify the strengths of the algorithm. The paper lacks the diversity of the data that must be improved. Evaluating one piece of data is not enough.
Moreover, the results are marginally equivalent to the existing algorithm. It is not easy to see the main contribution of this work. The author should list the main contribution of their work.
Moreover, adding dynamic convolution may increase the total number of parameter requirements; hence it may not be preferred over the other method with fewer parameters. Hence, some clarity is needed in this aspect.
The motivation is unclear, and dynamic convolution is needed in the case of Speech recognition. Further, how do you choose the dynamic window size?
The author also needs to provide a pseudo code of their algorithm.
The deep learning training and testing, and validation straggles need to be described much more straightforwardly.
The author should evaluate the performance of at least two other well-known datasets and compare them with the existing state-of-the-art.
Further, the ablation study shown in the paper does not give a complete picture of why DyCNN is helping. A better reason why and the importance of different modifications to existing architecture can be given.
Finally, is the transformer generally slow to train? Can you please elaborate on the training speed of your proposed modified architecture?
Highlight limitations of your method.
Reviewer 2 Report
The authors proposed a hybrid attention mechanism which combined the dynamic convolution CNNs and multi-head self-attention. The subject is interesting, but the results are not well-presented. They must be detailed to allow the correct comparison with further studies. There are some major comments:
- The references in the manuscript are not correctly presented;
- The introduction presented some background, but it must be improved with more recent references;
- The authors must present some related work to allow the discussion of the results obtained;
- The methods are well-presented, but please add some details (if possible) to allow the replication of the results;
- Most important, the results must be better presented and detailed with some charts and tables;
- The results must be discussed with the literature.